# Aprotinin (I): Understanding the Role of Host Proteases in COVID-19 and the Importance of Pharmacologically Regulating Their Function

**DOI:** 10.3390/ijms25147553

**Published:** 2024-07-10

**Authors:** Juan Fernando Padín, José Manuel Pérez-Ortiz, Francisco Javier Redondo-Calvo

**Affiliations:** 1Department of Medical Sciences, School of Medicine at Ciudad Real, University of Castilla-La Mancha, 13971 Ciudad Real, Spain; fernando.padin@uclm.es; 2Facultad HM de Ciencias de la Salud, Universidad Camilo José Cela, 28692 Madrid, Spain; 3Instituto de Investigación Sanitaria HM Hospitales, 28015 Madrid, Spain; 4Department of Anaesthesiology and Critical Care Medicine, University General Hospital, 13005 Ciudad Real, Spain; 5Translational Research Unit, University General Hospital and Research Institute of Castilla-La Mancha (IDISCAM), 13005 Ciudad Real, Spain

**Keywords:** proteases, aprotinin, COVID-19, kinin–kallikrein system (KKS), renin–angiotensin–aldosterone system (RAAS), angiotensin-converting enzyme type 2 (ACE2), thrombosis

## Abstract

Proteases are produced and released in the mucosal cells of the respiratory tract and have important physiological functions, for example, maintaining airway humidification to allow proper gas exchange. The infectious mechanism of severe acute respiratory syndrome coronavirus type 2 (SARS-CoV-2), which causes coronavirus disease 2019 (COVID-19), takes advantage of host proteases in two ways: to change the spatial conformation of the spike (S) protein via endoproteolysis (e.g., transmembrane serine protease type 2 (TMPRSS2)) and as a target to anchor to epithelial cells (e.g., angiotensin-converting enzyme 2 (ACE2)). This infectious process leads to an imbalance in the mucosa between the release and action of proteases versus regulation by anti-proteases, which contributes to the exacerbation of the inflammatory and prothrombotic response in COVID-19. In this article, we describe the most important proteases that are affected in COVID-19, and how their overactivation affects the three main physiological systems in which they participate: the complement system and the kinin–kallikrein system (KKS), which both form part of the contact system of innate immunity, and the renin–angiotensin–aldosterone system (RAAS). We aim to elucidate the pathophysiological bases of COVID-19 in the context of the imbalance between the action of proteases and anti-proteases to understand the mechanism of aprotinin action (a panprotease inhibitor). In a second-part review, titled “Aprotinin (II): Inhalational Administration for the Treatment of COVID-19 and Other Viral Conditions”, we explain in depth the pharmacodynamics, pharmacokinetics, toxicity, and use of aprotinin as an antiviral drug.

## 1. Introduction

Coronaviruses infect epithelial cells by recognising and binding to certain plasma membrane proteins. One of the most studied is the angiotensin-converting enzyme type 2 (ACE2). The anchoring mechanism of severe acute respiratory syndrome coronavirus type 2 (SARS-CoV-2), which causes coronavirus disease 2019 (COVID-19), to this enzyme has consequences that are fundamental for the cell and is closely related to the disease it causes. To understand the pathophysiology COVID-19, it is necessary to explain the importance of ACE2, which goes beyond blood pressure control. In this first review, titled “Aprotinin (I): Understanding the Role of Host Proteases in COVID-19 and the Importance of Pharmacologically Regulating Their Function”, we explain the pathophysiological importance of ACE in the SARS-CoV-2 infection process, and how a cascade of events activates proteolytic pathways that constitute the most important causes of the disease. Understanding these mechanisms allows the development of new antiviral drugs, such as aprotinin, which is a broad-spectrum inhibitor of host proteases. In the second-part review, titled “Aprotinin (II): Inhalational Administration for the Treatment of COVID-19 and Other Viral Conditions” [1], we describe the main pharmacodynamic, pharmacokinetic, and toxicological mechanisms of aprotinin for its use by inhalation in these conditions.

## 2. The SARS-CoV-2 Infectious Process

ACE2, the main target to which SARS-CoV-2 anchors in the viral infectious process, is expressed to a greater or lesser extent in the cells of the pulmonary, digestive, renal, and vascular endothelium [2,3]. Anchoring and fusion of the viral capsid to the host cell occurs through recognition of a virus envelope glycoprotein, known as the spike (S) protein, with this enzyme [4,5,6]. However, for the S protein to recognise it, it needs to undergo a post-translational activation process by endoproteolysis by proteases from the epithelial cells [7]. This two-step entry mechanism that involves activation by endogenous host proteases is common in viruses of the Paramyxoviridae, Orthomyxoviridae, Retroviridae, Herpesviridae, Flaviviridae, Filoviridae, Hepadnaviridae, Togaviridae, and Coronaviridae families [8,9]. In the specific case of SARS-CoV-2, the cleavage occurs in four redundant furin-like domains [10] located in the S1/S2 protein subunits of the virus [11]. Proteolysis is indispensable to separate and activate the S1 and S2 subunits, each of which performs distinct functions (Figure 1A) [12,13]. While the S1 subunit is responsible for binding to ACE2 with an affinity in the nanomolar range [14], the S2 subunit participates in the fusion between viral RNA and the cell membrane [13,15]. Unlike other coronaviruses, this proteolytic cleavage and presentation of the S1 subunit for anchoring to ACE2 is much more efficient for SARS-CoV-2 [16]. Therefore, it is essential to know the host proteases used by SARS-CoV-2 for its infectious process to prevent disease.

Among the known host proteases, trypsin, cysteine protease cathepsins, thermolysin, neutrophil elastase (NE), and activated clotting factors, such as plasminogen and factor Xa (FXa), have been described [13,17,18,19,20,21]. In addition, much attention has been paid to transmembrane serine protease type 2 (TMPRSS2), one of the proteases for which SARS-CoV-2 shows the strongest preference to infect epithelial cells [22,23]. Once anchorage to ACE2 occurs, and depending on which protease is used in this process, the virus can enter the cell via at least two mechanisms. If the S protein is cleaved via cathepsins, entry will occur through endosome formation. In contrast, if proteolysis is via TMPRSS2, entry occurs through the formation of a fusion pore between its membrane and those of the epithelial cell [24]. ACE2 is usually co-expressed with TMPRSS2, indicating the importance of this mechanism [25]. However, the endocytic pathway usually occurs when TMPRSS2 expression levels are insufficient, and endosome formation occurs via a clathrin-dependent pathway, internalising the virus bound to ACE2 [26]. The use of this endocytic pathway in respiratory cells that do not express TMPRSS2 explains why some SARS-CoV-2 variants, such as Omicron, have enhanced transmissibility [27,28].

Key points:SARS-CoV-2 uses both soluble and membrane proteases of the host cell in its viral infection mechanism.Depending on which protease you use will influence the way it enters the cell (e.g., endocytosis or through the formation of a fusion pore).

### 2.1. The Physiological Importance of ACE beyond Blood Pressure Control

Although ACE was initially described for its importance in the control of blood pressure through the renin–angiotensin–aldosterone system (RAAS) [29], it is now known to have relevance in other processes, such as renal embryonic development [30], reproduction [31,32], cell proliferation (e.g., haematopoiesis, myeloproliferation, and angiogenesis), inflammation, oxidative stress, and immunity [33,34,35]. In somatic tissues, ACE contains two catalytic domains, each with different affinities for protein substrates. While the carboxyl-terminal domain is specialised in cleaving angiotensin I to generate angiotensin II, the amino-terminal domain is also capable of cleaving other peptides, including enkephalin, the tetrapeptide N-acetyl-seryl-aspartyl-lysyl-proline (AcSDKP), neurotensins, substance P, and bradykinin, among others (Figure 2) [36]. It can exert its protease function while anchored to the plasma membrane, extracellularly when released from cells into the blood plasma or subcellularly in cell organelles. Thus, it is not only an endocrine, paracrine, and autocrine cellular communication system, it is also an intracellular communication system (between the cell organelles themselves) [37]. These locations are important for the development of its various cellular functions and are of great relevance for understanding COVID-19 as well as the mechanisms by which SARS-CoV-2 evades the immune system.

#### ACE in Immunity

ACE is expressed in organelles, such as the endoplasmic reticulum, where it has been shown to be catalytically active [38], performing certain cellular functions. Among these, it contributes to adaptive immunity by cleaving both endogenous and foreign peptides for presentation by antigen-presenting cells (macrophages and dendritic cells) via major histocompatibility complex type I (MHC-I) and II (MHC-II) to cluster of differentiation 8 (CD8)^+^ T cells (Figure 2) [38,39,40].

In addition, ACE can be overexpressed in myeloid cells, such as macrophages, neutrophils, and dendritic cells, thereby enhancing the innate immune response through the production of proinflammatory cytokines and nitric oxide [41]. ACE plays a role in modulating the inflammatory response and recruitment of inflammatory cells, such as mast cells and neutrophils, by regulating the concentration of mediators and/or the expression of proteins, such as cytokines, adhesion molecules, and the plasma contact system, also known as the kinin–kallikrein system (KKS). ACE regulates this system by clearing the proinflammatory peptide bradykinin [42]. Increased bradykinin contributes to the activation of the FXII coagulation and complement pathways [43]. ACE can be secreted into the extracellular environment (soluble ACE (sACE)) from these myeloid cells and act as a local or systemic regulator of the production of these peptides [41]. In neutrophils, ACE is essential for immune competence [44]. These effects are independent of the angiotensin-converting action and depend on other factors—for example, activation of nicotinamide adenine dinucleotide phosphate oxidase (NOX) by ACE increases reactive oxygen species (ROS, including the superoxide) production [41,44]. Neutrophils kill bacteria using phagocytic mechanisms and through the release of extracellular fibres, called neutrophil extracellular traps (NETs), which are in turn stimulated by ROS generation. These fibres are composed of DNA and proteins that bind to, trap, and consequently kill bacteria. Moreover, under certain conditions, monocytes overexpress ACE, triggering further differentiation of these cells into macrophages, which release cytokines and adhesion and transmigration molecules [45]. In addition, ACE is also involved in the production of nitric oxide, which is important for microbial defence by these cells (Figure 2) [46].

Finally, ACE is involved in the control of immunity through the clearance of peptides with immunosuppressive activity, such as AcSDKP [36,47]. Among many actions, this peptide can inhibit the G1–S cell cycle transition and thus maintains haematopoietic progenitor cells in a quiescent phase [48]. This prevents proliferation, migration, and cytokine release by myeloid cells [49,50]. In addition, it can prevent fibroblast collagen synthesis and deposition by inhibiting DNA synthesis and endothelin-1, and by blocking small mothers against decapentaplegic (Smad) signal transduction and the extracellular signal-regulated kinase 1/2 (ERK1/2) pathway, thus preventing the action of cytokines, such as transforming growth factor-β (TGF-β) [51]. AcSDKP has antifibrotic and anti-inflammatory effects in the lung, heart, liver, and kidney [51,52]. Therefore, ACE may exert these effects by decreasing angiotensin II levels and increasing AcSDKP levels [50].

Key points:ACE cleaves a multitude of peptides acting subcellularly, anchored to the plasma membrane, or extracellularly.ACE is involved in cleaving peptides to be presented by antigen-presenting cells at MHCs. Its dysregulation may contribute to viral evasion mechanisms.ACE is overexpressed in myeloid cells as a mechanism of immune response and inflammation.ACE regulates the contact system and the KKS.ACE regulates immunity through the clearance of peptides, such as Ac-SDKP.

## 3. The Protease and Anti-Protease System in the Lung and SARS-CoV-2 Infection

The respiratory epithelium is mucosal tissue that consists of more than 40 different cell types [53]. These cells coordinate their actions through the release of various mediators (mucins, cytokines, proteases, and anti-proteases) to maintain respiratory function and protect against foreign agents. Proteases that are released from the respiratory mucosa are involved in processes related to airway function, including mucus characteristics (i.e., its density or rheology) [54], mucociliary clearance [55,56], and the recruitment and function of immune cells [57,58]. This process is carried out by cleaving pre-proteins to their physiologically active forms, a process that is finely regulated by the action of anti-proteases to control excessive over-activation [59]. The human genome contains more than 600 proteases, which gives an idea of their importance [60].

When infection occurs, the respiratory epithelium starts to produce proteases, such as the type II transmembrane serine proteases (TTSPs). This family includes matrix metalloproteinases (MMPs), a disintegrin and metalloproteinases (ADAMs), a disintegrin and metalloproteinase with thrombospondin motifs (ADAMTSs), cathepsins, proteinase-3, human neutrophil elastase, trypsin, chymotrypsin, prostasin, and TMPRSSs. In addition to the aforementioned processes, these proteases regulate tissue repair, blood coagulation (secondary haemostasis), fibrinolysis, and immune function [61]. Moreover, anti-proteases or serpins can be released from respiratory epithelia, including α-1-antitrypsin (serpin A1), plasminogen activator inhibitor 1 (serpin E1), and glia-derived nexin (serpin E2), which can reduce SARS-CoV-2 infection [62]. The underlying pulmonary pathological processes involve an imbalance between proteases and anti-proteases [63], and viruses are known to deregulate this balance [64,65,66,67]. Proteomics studies in cells infected in vitro with SARS-CoV-2 have shown dysregulation with increased expression of proteases and decreased expression of anti-proteases (e.g., SPINT1 (Kunitz-type protease inhibitor 1), SPINT2 (Kunitz-type protease inhibitor 2), tissue inhibitors of metalloproteinases 1/2 (TIMP1/2), amyloid beta precursor-like protein 2 (APLP2), cystatin C (CST3), and α-1-antitrypsin) [68,69]. Therefore, it is necessary to re-establish their balance to have a healthy mucosa.

Key points:The respiratory mucosa controls its functions through a complex release system of proteases and anti-proteases.Respiratory viruses in their infective process unbalance the action of proteases and anti-proteases.

### 3.1. Host Proteases in SARS-CoV-2 Infection

Coronaviruses, such as SARS-CoV-2, take advantage of the activity of multiple proteases to infect a host cell [21]. When respiratory epithelial cells become infected, hyperreactivity occurs, where proteases (e.g., trypsin, cathepsins, and/or elastases) are overexpressed and released from respiratory tract epithelial cells as well as myeloid cells. Excessive activity of these proteases significantly contributes to the inflammatory and/or infectious processes. We have already discussed the role of TTSPs, such as TMPRSS2, in the process of SARS-CoV-2 entry into a host epithelial cell [23]. Below, we briefly review other proteases that also play an important role in the pathological process in the respiratory tract.

#### 3.1.1. ADAMs

ADAMs are a family of type I transmembrane proteins belonging to the adamalysin subfamily of metalloproteinases. The members of this family have a metalloprotease domain and an integrin-interacting domain (disintegrin domain), indicating that they have both protease and adhesion molecule activity. They participate in the cellular processes of migration, adhesion, and cell fusion. In addition, through their protease activity, they participate in cell signalling by cleaving certain protein domains (sheddase activity), leading to the release of cell membrane-associated proteins, such as cytokines, apoptotic ligands, growth factors, and receptors. One of the most studied examples is the production of TNF-α. ADAM17 is also known as the TNF-α convertase (TACE)—it cleaves transmembrane TNF (26 kDa) to release its active soluble form, TNF-α (17 kDa). This family of proteases also produce mucus at the bronchial level [70]. In COVID-19, ADAM17 is upregulated via internalisation of ACE2 after it binds to SARS-CoV-2, a phenomenon that has been observed with other coronaviruses [71]. In addition, ADAM17 proteolytically cleaves ACE2, releasing it in its soluble form (sACE2) and decreasing its expression in the cytoplasmic membrane [72]. This phenomenon contributes to the infectious process [73] as well as the release of TNF-α, interleukin 6 (IL-6), and other proinflammatory molecules, which aggravates the inflammatory process (Figure 1B) [73,74].

#### 3.1.2. Elastases and Other Neutrophil Serine Proteases

Elastases are serine proteases that have important physiological functions through the cleavage of multiple protein substrates. In coronavirus infections, elastases, such as pancreatic elastase, are involved in cleavage of the S protein [75,76] via an elastase-specific domain in the S2 subunit [75]. For SARS-CoV-2, the involvement of elastases released from myeloid cells such as macrophages [77], but mostly neutrophils, has been investigated. Their importance lies in the viral entry process [78]. In addition, the release of human neutrophil elastase from neutrophils is associated with increased production of cathepsins and matrix metalloproteases [79], which promote infection. Along with human neutrophil elastase, neutrophils release other serine proteases, such as cathepsin G, proteinase-3, neutrophil serine protease-4 (NSP-4), azurocidin (AZU1), myeloperoxidase, myeloblastin (PRTN3), and transcobalamin-1 (TCN1), which are stored in azurophil granules [80]. The main functions are to degrade the extracellular matrix by digesting collagen, transmembrane proteins, pulmonary surfactant factors, and proteoglycans. Therefore, their release contributes to a process of permeabilisation of cellular barriers, inflammation, and alteration of lung functions. In addition, they participate in inflammation by processing and modifying cytokine functions, inhibiting anti-inflammatory factors, such as progranulin, activating surface receptors, such as Toll-like receptors (TLRs) through their protease action, and promoting cytokine release from monocytes and macrophages [80,81]. However, in COVID-19, the most prominent role of human neutrophil elastase is in NET formation and NETosis, a process in which neutrophils expel their citrullinated chromatin into the extracellular milieu. Combined with the action of released proteases, a network is produced with the capacity to trap platelets and red blood cells, linking the thromboembolic processes that occur during COVID-19 [82]. The exacerbation of these processes is partially caused by an imbalance between the release and activity of proteases secreted by neutrophils, as opposed to anti-proteases, such as α-1-antitrypsin and other inhibitors secreted from leucocytes [78].

#### 3.1.3. Trypsin and Human Airway Trypsin-like Proteases (HATs)

Trypsin has been shown to increase the infectivity of several coronaviruses, including SARS-CoV-2, when the virus is already attached to its cellular target [13,76,83]. These mechanisms do not necessarily require binding to ACE2, but they favour increased membrane fusion. In certain SARS-CoV-2 variants (e.g., Delta) that have a greater capacity to produce syncytia [84], and therefore have greater membrane fusion activity, trypsin plays a key role in this infectious mechanism [85]. On the other hand, excess trypsin release contributes to cell barrier permeabilisation, inflammation, and impaired lung function [86].

HATs are preferentially anchored to the surface of bronchial and tracheal respiratory tract hair cells [87]. They can also be found in soluble forms in patients with respiratory diseases [88]. Coronaviruses that need to cleave proteins from their capsid during the infectious process can use these proteases [89]. In addition, they participate in the activation of epithelial sodium channels (ENaC) that hydrate the airway and facilitate mucociliary clearance [90]. ENaC and the coronavirus S protein share the same furin-like cleavage domain, and thus use the same proteases (e.g., TMPRSS2) for protein activation [91]. Hence, by modifying the activity of proteases after infection, coronaviruses may affect ENaC activity [91,92]. SARS-CoV-2 leads to the overexpression of proteases, such as TMPRSS2, at the host plasma membrane and simultaneously prevents these proteases from degrading ENaC, leaving them in an overactivated state [93]. This state accelerates viral entry into the host cell [94]. In addition, altered ENaC activity is consistent with symptoms following SARS-CoV-2 infection, such as a runny nose, ageusia, pulmonary oedema, and respiratory distress [91].

The increased release or expression of proteases, such as trypsin, HATs, and others secreted by neutrophils (e.g., elastase and proteinase-3), favours activation of protease-activated receptor type 2. This receptor promotes the release of inflammatory mediators [95,96,97] that aggravate the inflammatory process [98] and cell growth, contributing to airway remodelling through fibroblast proliferation [99]. In addition, they can cleave and activate the urokinase-type plasminogen-activated receptor [100], which also contributes to thromboembolic processes in patients with COVID-19 [98].

#### 3.1.4. Cathepsins

Cysteine cathepsins are a family of papain-like proteases found intracellularly in organelles such as lysosomes, although they have also been observed in the cytosol, mitochondria, nucleus, plasma membranes, and the extracellular milieu [101,102]. Cathepsin secretion into the extracellular milieu is observed under physiological conditions; for example, cathepsin B, K, and L are secreted from thyroid epithelial cells to release thyroid hormones from thyroglobulins. However, excessive cathepsin release can also be triggered in pathological conditions that involve inflammation [102,103]. In infectious diseases, such as COVID-19, where inflammation is evident, their expression and secretion increase [80,104]. A relevant fact is the ability of cathepsins to modify the spatial conformation of proteins by cleaving certain domains to alter their functionality [102]. The high transmissibility of SARS-CoV-2 variants, such as Omicron, has been linked to a greater capacity to enter the cell via an endocytic pathway, thanks to cathepsin-mediated modification of the S protein to produce the correct spatial arrangement of the S2 subunit [24,27,28,105]. A large number of cathepsin subtypes have been proposed to be involved in SARS-CoV-2 infection [18,21]. Researchers have used in silico and in vitro approaches to determine the exact S protein cleavage sites [18,106]. Cathepsins are upregulated upon infection, which contributes to aggravating infection, especially in those tissues where other proteases, such as TMPRSS2, are not as present, thus allowing a new entry route to infect these cells [68,80]. Another relevant factor regarding the action of cathepsins in the extracellular milieu is that they are important initiators or suppressors of the activity of other proteases or cytogens involved in proteolytic cascades. For example, cathepsin G is one of the main proteases, along with human neutrophil elastase, involved in NET formation, inflammation, and thrombosis. The mechanisms involve inactivation of tissue factor inhibitory peptide and activation of protease-activated receptors [107]. In addition, cathepsins are potent activators of blood platelets [108], and cathepsin L activates the KKS [109].

Cathepsin F, L, S, and V are also present in endolysosomes, where they cleave proteins and contribute to maturation and processing functions. An example of these functions is the processing of ENaC subunits [110]. The presence of SARS-CoV-2 and its viral proteins in lysosomes interferes in the proteolytic cleavage of ENaC and in its maturation processes that alter its functionality. These changes contribute to COVID-19 symptoms, as ENaC is involved in proper homeostasis of the pulmonary fluid interface [91] and in the occurrence of electrolyte imbalance (i.e., hypokalaemia) due to altered renal function [93,111]. Another example is the processing of antigenic proteins in immune cells for presentation by MHC-I and MHC-II [112]. Moreover, lysosomal cathepsins influence the trafficking of SARS-CoV-2-specific proteins, such as the accessory protein open reading frame 3a (ORF3a), which is involved in virus infectivity and the formation of new virions [96,113]. Indeed, if cathepsins are inhibited, it can lead to degradation of new virions into cellular multivesicular bodies [114].

SARS-CoV-2 may use a lysosomal pathway that involves cathepsins to release newly formed virions from the host cell [115], although the mechanisms remain unclear. In addition, cathepsin L is involved in the upregulation and processing of heparanase, which is implicated in the release of viral progeny and their propagation [116,117,118]. Serum heparanase levels are increased in patients with COVID-19 and correlate with the severity of the disease [116,119]. Specifically, heparanase damages the glycocalyx of endothelial cells and promotes thromboembolism [120]. It causes the release of certain molecules from heparan sulphate proteoglycans of the glycocalyx that bind to it, including growth factors, cytokines, enzymes, and lipoproteins [121]. These molecules released by heparanase are involved in cell motility, angiogenesis, inflammation, coagulation, stimulation of autophagy, and exosome production [122,123].

Since cathepsins are involved in the mechanisms of entry, processing of viral proteins, and release of viral progeny to the cellular exterior, their inhibition is a very attractive and well-studied therapeutic strategy for the development of antiviral drugs against coronaviruses [124,125].

Key points:SARS-CoV-2 infection upregulates ADAM-17, causing the adhesion of inflammatory cells and the release of chemoattractants, such as sACE2 and proinflammatory cytokines (IL-6 and TNF-α).Elastases participate in viral entry into the host cell and the release of other proteases that amplify the infectious and inflammatory response, permeabilise cellular barriers, and alter lung functions. Neutrophil elastase participates in NETosis.Trypsin promotes infection when the virus is attached to the cell membrane by increasing the membrane fusion process.Coronaviruses prevent proteases from exercising their physiological functions of correct maturation of proteins, such as ENaC. This is involved in many of the symptoms caused by respiratory viruses.Cathepsins allow the entry of SARS-CoV-2 via the endocytic route into those cells where other proteases, such as TMPRSS2, are not so present.Cathepsins initiate and amplify the activation of pathways activated by proteases such as KKS, coagulation, or the formation of NETs.Cathepsins participate in entry, maturation of viral proteins, and release of new viral progeny from the host cell.

### 3.2. Host Anti-Proteases in SARS-CoV-2 Infection

#### 3.2.1. α-1-Antitrypsin (Serpin A1)

In addition to increasing protease activity, coronaviruses negatively modulate the action of anti-proteases, which contributes to an imbalance between proteases and anti-proteases that prevents proper respiratory function. In general, coronaviruses increase the degradation of α-1-antitrypsin (serpin A1), increasing its degradation products. This mechanism is associated with increased pathogenicity [126,127] because α-1-anthrypsin is one of the most important inhibitors of serine proteases in the lung and, therefore, has important anti-inflammatory actions [128]. The loss of α-1-anthrypsin activity is also relevant to the SARS-CoV-2 infectious process [126]. For example, α-1-antitrypsin inhibits TMPRSS2, which is required for S protein cleavage [129], thus aggravating the infection. Once infection has occurred, the inflammatory process starts, with IL-6 serving as one of the main mediators and a prognostic marker of the disease [130,131]. One of the main actions of IL-6 is to induce hepatic synthesis of α-1-antitrypsin. It does so through the formation of IL-6-soluble IL-6 receptor (sIL-6R) complexes and subsequent binding to glycoprotein 130 (gp130), which activates the Janus kinase (JAK)/signal transducer and activator of transcription (STAT), ERK, and phosphoinositide 3-kinase (PI3K) signalling pathways in liver cells, regulating gene transcription [132]. As we discuss later, this is due to an increase in the release of sIL-6R via the action of proteases, such as ADAM17 [73,133], as opposed to action on un-cleaved membrane receptors released by proteases. Hence, an increased IL-6/ α-1-antitrypsin ratio has been proposed as a biomarker for a poor prognosis in SARS-CoV-2 infection [130,131]. The fact that anti-proteases, such as α-1-antitrypsin—whose function is to inhibit proteases (e.g., human neutrophil elastase, cathepsins, and metalloproteases) that are activated when the infectious process occurs, as a regulatory control mechanism [128]—lose functionality results in infection and inflammation. In addition, it is involved in the regulation of coagulation by inhibiting thrombin [134]. Thus, patients with COVID-19 and attenuated α-1-antitrypsin production tend to have a poor prognosis [130,131]. Therefore, α-1-antitrypsin is being evaluated for its therapeutic utility in several trials for the treatment of COVID-19 (ClinicalTrials.gov IDs: NCT04385836, NCT04547140, NCT04495101, and NCT04817332 [134,135]).

#### 3.2.2. CST3

Cystatins are a superfamily of proteins that are ubiquitously expressed in all nucleated cells and consist of at least one 100–120 amino acid inhibitory domain with protease activity. There are three types of cystatins: type I or Stefins, which are cytosolic proteins, type II, which are secreted from cells into the extracellular milieu, and type III or quininogens, which are multifunctional proteins found in the blood and other fluids [136]. CST3 is secreted in body fluids, such as saliva and urine [137]. It has antiviral activity against coronaviruses and, therefore, its recombinant forms have been proposed as antiviral drugs against them [138,139]. The antiviral mechanism is related to the ability to inhibit cysteine proteases, such as cathepsins, which are used by the virus to gain entry to a host cell via the endocytic pathway [136,139]. Cathepsin S and L, among others, have an important role in foreign protein digestion, processing, and loading onto MHC-I and MHC-II for presentation by antigen-presenting cells (including dendritic cells) to T lymphocytes. By inhibiting cathepsins, cystatins regulate the generation of peptidergic MHC antigenic complexes [140]. This process involves other related proteins that have the capacity to inhibit cathepsins. The p41 isoform of CD74 of MHC-II has recently been found to inhibit them, in addition to its antigen-presenting function. In turn, CD74 can be activated through the MHC-II master regulator MHC-II transactivator (CIITA) [141]. Interestingly, CIITA levels have been observed to decrease in both children and adults when the prognosis of COVID-19 is severe [142]. The importance of this chain of events—ranging from CIITA activation of p41 of CD74 to cathepsins as final effector proteins—is that in addition to participating in antigen presentation, they intervene in the antiviral response via interferon (IFN) [143]. Finally, inhibition of cathepsins causes virions to be redirected to a degradation pathway in multivesicular bodies or lysosomes. Hence, inactivation of these pathways plays a very important protective role via reduction of viral transcription, assembly, and release [144].

Paradoxically, many studies on COVID-19 have reported a positive correlation between elevated serum and urine CST3 levels and mortality [145,146,147]. CST3 is a highly sensitive biomarker in the assessment of cardiovascular and renal function because it is filtered only at the glomerular level. Of note, renal failure is one of the main causes of mortality in patients with COVID-19 [148,149]. On the other hand, in the saliva of patients with moderate or severe COVID-19, CST3 levels have been observed to be decrease with respect to healthy individuals [150,151,152]. In hospitalised patients, CST3 levels tend to increase in those who are symptomatic relative to those who are asymptomatic, although these values are always lower than in healthy individuals [150]. These differences in CST3 saliva and serum levels are most likely due to renal impairment and the fact that many of these patients are treated with glucocorticoids that increase CST3 plasma levels [153]. In addition, elevated levels of ILs, such as IL-6, downregulate cystatin levels, and vice versa [154]. Critically ill patients with COVID-19 have high IL-6 levels [131] and show an abnormal glomerular filtration rate (GFR). Specifically, the GFRcystatinC/eGFRcreatinine ratio is <0.6 due to the influence that cytokines, such as IL-6, have on the regulation of CST3, among other factors [155,156].

#### 3.2.3. Other Anti-Proteases

Other anti-proteases that are affected after SARS-CoV-2 infection include secretory leucocyte protease inhibitor (SLPI) and elafin [157]. SLPI is one of the most important anti-proteases secreted by clear and goblet epithelial cells, submucosal glands, and leucocytes in the airways as a protective mechanism against damage [158]. In COVID-19, SLPI is released as an anti-inflammatory response after cytokine release [157]. Factors such as pulmonary surfactant A released from type II pneumocytes may contribute to its release, and infection of these cells by SARS-CoV-2 may impair these mechanisms [159,160]. In patients with COVID-19, although increased expression has been observed [157,161], it is not able to attenuate the action of proteases, such as human neutrophil elastase [157]. In COVID-19, there is an increased release of proteases, such as MMP-9 and/or elastase released from neutrophils, which may contribute to the clearance of SLPI. In severe cases of COVID-19, this intense protease activity leads to increased oxidative stress, which promotes neutrophil activation. This leads to the suppression of pathways, such as nuclear factor erythroid 2-related factor 2 (Nrf2), to poise this oxidative balance [162,163] and, consequently, an imbalance in anti-protease activity. The administration of SLPI in its recombinant form has been proposed as a therapeutic alternative to treat COVID-19 [164].

Key points:Coronaviruses increase the degradation of anti-proteases, such as α-1-antitrypsin.Cystatin is one of the main anti-proteases that regulate the action of cathepsins.

## 4. Consequences of SARS-CoV-2 Anchoring to ACE2

Cells of the respiratory epithelium are targets for more than 200 viruses, which infect them through recognition and anchoring to proteins on their plasma membrane [165]. Several binding proteins for SARS-CoV-2 have been proposed, including neurophilin-1 [166], metabotropic glutamate receptor subtype 2 (mGluR2) [167], kidney injury molecule-1 [168], heat shock protein A5 (HSPA5) or glucose-regulated protein-78 (GRP78) [169], basigin (CD147) [170], heparan sulphate [171], dipeptidyl peptidase 4/CD26 [172], GRP78, CD147 [173], and TLR4 [174]. However, ACE2 is considered the most important target [2,3] because it is the one that can best explain the pathological events that occur after infection. Its expression levels in different tissues predict the main target organs of SARS-CoV-2: digestive tract (ileum) > heart > kidney and urinary bladder > respiratory tract and lung >>> stomach and liver [175]. Specifically, ACE2 expression is much higher in alveolar tissue (type II pneumocytes) relative to the other parts of the respiratory tract, explaining the great damage the SARS-CoV-2 infection causes there [176].

With the entry of the virus through the airway, and the subsequent proteolytic cleavage of the S protein by host proteases, the S1 subunit interacts with ACE2 exposed on the surface of mucosal cells [4,15,16,23,25]. This anchoring facilitates the entry of the virus into epithelial cells. The mechanism involves internalisation of ACE2 and its degradation by a protein that regulates cholesterol: protein convertase subtilisin-kexin type 9 (PCSK9) [177]. Therefore, there is a reduction in ACE2 expression on the surface of the membranes of infected cells [178,179]. This reduction is not only due to internalisation and degradation. In addition, coronaviruses downregulate ACE2 mRNA levels in tissues, such as the lung and myocardium [180,181]. ACE2 expression levels in the lung have been associated with a protective factor against respiratory distress [182]. A partial explanation for ACE2 upregulation caused by the virus is linked to the overexpression of ADAM17 via a mechanism that is still unclear [71]. ADAM17 is involved in the removal of ACE2 ectodomains from the cytoplasmic membrane, which consequently leads to an increase in free sACE2 (Figure 1B) [73,183,184]. In addition, it also causes the release from the plasma membrane of proinflammatory factors, such as TNF-α and its receptors TNF receptor 1 and 2 [185,186]. The same is true for IL-6R: the release of sIL-6R and the formation of IL6–sIL-6R complexes, which in turn bind to gp130 that is expressed on many cell membranes, exacerbates IL-6 production through the JAK/STAT3 pathway [133,187]. Autopsy studies of patients who died from coronavirus infections revealed that ACE2^+^ cells, which are infected by the virus, produce the most proinflammatory cytokines [188]. Furthermore, increased accumulation of sACE2 throughout the infectious process correlates independently with mortality in SARS-CoV-2 infection [183,189,190,191]. Thus, the accumulation of sACE2 is a measure by ADAM17 activity and is related to the inflammatory process. On the other hand, ACE2 downregulation leads to an imbalance in the production of angiotensinogen-derived peptides and, consequently, an imbalance between the angiotensin II/angiotensin II receptor type 1 (AT1R) and its counterpart angiotensin 1–7/MAS receptor pathways [190,192]. In addition, ACE2 is responsible for the degradation of other peptides, such as apelin (an APJ receptor agonist) and des-Arg^9^-bradykinin (a bradykinin receptor B1 agonist), linking this protease to the KKS [193,194]. We will discuss the importance of bradykinin in COVID-19 later.

One of the main consequences of ACE2 downregulation is increased production of angiotensin II, which is closely correlated with the viral load [195]. Angiotensin II acts on the AT1R, which has been implicated in the control of blood pressure and electrolyte balance. However, after SARS-CoV-2 infection, AT1R exacerbates vasoconstriction, inflammation, cell proliferation, fibrosis, thrombosis, and oxidative stress via ROS production [196]. AT1R is expressed on myeloid cells (dendritic cells and macrophages), neutrophils, mononuclear cells, T and B lymphocytes, and non-immune tissue cells. The latter are the most reactive [133,197]. Their activation results in the release of inflammatory mediators, such as vascular endothelial growth factor (VEGF), prostaglandins, TNF-α, IL-1β, IL-6, IL-10, and ROS [133,198]. These actions are mediated by multiple signalling pathways, including NOX, NF-κB, ERK1/2, MAPK, and STAT1 [199,200]. These signalling pathways are the main molecular players in triggering a hyperinflammatory state (cytokine storm) and acute respiratory distress syndrome [196]. These mechanisms provide insight into why angiotensin II leads to apoptosis of the alveolar epithelium through AT1R [201].

Key points:ACE2 is the most important entry target of SARS-CoV-2 since the dysfunction in its activity caused by the infection best explains the COVID-19 disease.Viral entry causes an overexpression of ADAM17, a decrease in ACE2 in the plasma membrane, and an increase in its soluble forms, which causes inflammation due to the release of TNF-α and IL-6.Dysregulation of ACE2 causes an increase in angiotensin II that contributes to the hyperinflammatory state, respiratory distress, and damage to the lung epithelium.

### 4.1. Dysregulation of ACE2 and Its Relation to Bradykinin

Kinins, including bradykinin, require the action of plasma or tissue kallikrein for their synthesis [202]. In the plasma, pre-kallikrein (Fletcher factor) is cleaved, transforming it to an enzyme with serine-protease activity. Kallikrein cleaves a pre-protein kininogen to produce high-molecular-weight kininogen (HMWK; also known as Fitzgerald factor) and bradykinin [203,204]. Tissue-derived kallikrein has different properties than plasma kallikrein [202]. It is released physiologically at lower concentrations, mainly from exocrine glands, such as the salivary glands, kidney, and pancreas [205]. In addition, it can be synthesised by the activity of polycarboxypeptidase, also known as angiotensinase C, which is constitutively expressed on the surface of endothelial cells [206]. When activated (e.g., by pathogens), this enzyme converts plasma pre-kallikrein to kallikrein. Kininogens can also be cleaved by serine proteases other than kallikreins, such as human neutrophil elastase, tryptase, cathepsins, and proteinase-3 [207,208]. These processes also increase the production of kinins, such as Lys-bradykinin from L- and H- kininogens, which are rapidly bio-transformed to bradykinin via plasma aminopeptidases [202,209,210].

One of the main functions of bradykinin is to participate in inflammation, pain, and innate immunity [211]. Kallikreins are involved in important physiological functions, such as activation of the intrinsic pathway of blood coagulation, its regulation, and fibrinolysis. In addition, bradykinin, HMWK, and FXII are involved in the contact system of innate immunity, which in turn participates in activation of the complement pathway (Figure 3) [211,212,213].

The KKS is considered an extension of the RAAS [214]. ACE has a greater affinity for bradykinin than for angiotensin II [215]. Thus, after kinin production, ACE2 preferentially degrades des-Arg^9^-bradykinin, whereas ACE preferentially metabolises bradykinin [193,194,216]. Considering the rapid activity of ACE, the half-life of bradykinin and Lys-bradykinin is only 27 s, so these proteins act locally. In addition, ACE bio-transforms 11% of bradykinin to des-Arg^9^-bradykinin, which increases its half-life ten-fold compared with bradykinin [216]. The different ACE isoforms, which have distinct activity, allow bradykinin to regulate angiotensin II activity and promote vasodilation, natriuresis, and hypotension through the bradykinin receptor B2 [217]. These ACE isoforms exert peptidylpeptidase and kinase II activities and can metabolise kinins and kallikreins (Figure 3) [218,219]. Furthermore, angiotensin II increases B2 receptor expression [220,221,222], and angiotensin II receptor type 2 stimulates the expression of angiotensinase C in endothelial cells and, thus, the production of bradykinin [223].

Regulation of the expression and activity of ACE and its isoforms is crucial. When ACE2 is downregulated, there will be a preferential increase in des-Arg^9^-bradykinin levels [224,225]. As mentioned previously, ACE2 is downregulated in patients with COVID-19 [178,179,183,190,191]. An increase in the ACE/ACE2 expression ratio has been associated with organ damage in these patients [224]. Moreover, Roche and Roche [226] suggested that plasma kinin concentrations should be monitored to help predict the severity of pulmonary problems. In addition, kinins have been implicated in pain and inflammation that is potentiated by plasmin [43]. Des-Arg^9^-bradykinin activates the B1 receptor with great affinity, leading to neutrophil recruitment, increased vascular permeability, and leucocyte extravasation into the lung [217]. Activation of this receptor leads to the release of chemokines, such as C-X-C motif chemokine 5 (CXCL5), from pulmonary epithelial cells, which in turn activates chemokine receptor type 2 (CXCR2) of neutrophils to facilitate their recruitment and infiltration into the lung and increasing inflammation [227]. In these inflammatory conditions, the B1 receptor is upregulated almost 3000-fold and the B2 receptor is upregulated 200-fold compared with the normal state, resulting in respiratory distress syndrome and multiorgan failure [216,224,225,228]. B1 and B2 receptor activity is highly sensitive to the action of cytokines and growth factors (e.g., IL-1β, IL-2, TNF-α, IFN-γ, and epidermal growth factor (EGF)) and toxins from microorganisms [216]. These effects, observed in patients with COVID-19, have also been detected in murine models with ACE2 downregulation. Furthermore, in these models, ACE2 downregulation exacerbates inflammation and pulmonary oedema via increased B1 receptor activation by elevated Des-Arg^9^-bradykinin levels [229]. The B2 receptor has also been linked to the production of fever, cough, bronchoconstriction, and increased airway resistance [216,230]. It is also important to note that while plasma kininogen and kallikrein concentrations are very low in healthy individuals, very high values are detected in patients with COVID-19 [216,225,228,231].

Bradykinin also increases the activity of chymase [232], an endopeptidase released by mast cells to increase angiotensin II release and thus enhance the inflammatory process [233]. Bradykinin increases renin synthesis and release through the B2 receptor and induction of prostaglandin E2 release [234]. In addition, tissue kallikreins can transform angiotensin I to angiotensin II [235]. In summary, bradykinin is substantially involved in the cytokine storm that occurs in patients with COVID-19 [225,230,236,237] and in the various symptoms that occur during the course of this disease.

Key points:Bradykinin, HMWK, and FXII of coagulation form the backbone of the contact system of innate immunity and participate in complement activation.The KKS is an extension of the RAAS. ACE has peptidylpeptidase and chymase II activity, so it not only metabolises angiotensin II but also kinases and kallikreins.The imbalance of the ACE/ACE2 ratio is associated with an increase in des-Arg^9^-bradykinin, which is related to neutrophilia, inflammation, and increased lung tissue damage.Des-Arg^9^-bradykinin is related to many of the symptomatic processes of COVID-19 (fever, cough, or bronchoconstriction).By increasing chymase activity, bradykinin causes higher levels of angiotensin.

### 4.2. Involvement of the RAAS and the KKS in Thromboembolism

One of the most frequent causes of mortality after SARS-CoV-2 infection is the formation of micro-embolism and macro-embolism in the pulmonary and extrapulmonary vasculature [238]. The causes of these events are related to vascular endothelial and epithelial cellular dysfunction [239] that occurs after ACE2 internalisation. The phenomenon is related to three fundamental processes that occur after infection: (**a**) immune activation and production of proinflammatory cytokines by endothelial cells, (**b**) dysregulation of the RAAS, and (**c**) dysregulation of the KKS. In this section, we mainly discuss how RAAS and KKS dysregulation affect the thromboembolic processes.

#### 4.2.1. Immune Activation in Thromboembolism

Patients with COVID-19 present an exacerbated inflammatory response, known as the cytokine storm [240]. Epithelial cells (especially pulmonary and vascular cells) play an important role in this phenomenon, leading to tissue damage and immuno-thrombosis. This syndrome, which has a high incidence in patients with COVID-19 (10–20%), is characterised by high morbidity (e.g., micro-embolism, macro-embolism, and/or multiorgan failure) and lethality [240]. This situation is caused by expression of the viral proteins open reading frame 3b (ORF3b), ORF6 and ORF8 in infected cells. Together with the nucleocapsid (N) protein, these proteins are involved in facilitating rapid viral replication in epithelial cells (e.g., the vascular endothelium) and delaying or suppressing the response to type I and II IFNs, a process that involves NF-κB [241]. Furthermore, ACE2 dysregulation occurs as a consequence of viral entry and the important roles that this enzyme has in immunity—including control of immune competence of myeloid cells and the clearance of peptides, such as AcSDKP, involved in activation of immunity [36,47], leading to dysregulated activity of CD8^+^ T lymphocytes, natural killer (NK) cells, and antigen-presenting cells. This causes a miscommunication between innate and adaptive immunity, inducing amplification of the cytokine-mediated inflammatory response, which is prolonged over time [242].

SARS-CoV-2 infection of endothelial cells [243] causes dysfunction either directly, via viral activation of signalling pathways (e.g., the S protein binds to ACE2 to cause calcium-dependent toxic effects in endothelial cells [244]), or indirectly by altering the endothelium-associated immune and inflammatory response [146,245]. This inappropriate response is also aggravated by infection of vascular basement membrane pericytes [245,246], which have high ACE2 expression [245,247]. This leads to an imbalance in the endothelium–pericyte relationship, which has consequences in the signalling and the release of proinflammatory and profibrotic factors, such as angiopoietin I or platelet-derived growth factor [161,248]. The combination of the correct production of these mediators and the degree of expression of their receptors, as well as pericyte death [249], triggers hypercoagulation, vascular permeability, oxidative stress, and the passage of toxins into adjacent tissues. This phenomenon has also been observed with SARS-CoV-2 [250], as demonstrated in autopsies of patients with COVID-19 [251].

SARS-CoV-2 also activates complement pathways [252,253,254], which are closely and reciprocally related to haemostasis. This activation may be due in part to endothelial damage and to the action of several SARS-CoV-2 structural proteins. The N protein enhances the activation of the complement lectin pathway [255], and the S protein activates the alternative complement pathway by binding to heparan sulphate on cell surfaces and to C4a [256,257]. Both complement pathways converge with activation of C3a and C5a [252,253], anaphylatoxins that increase immune cell recruitment and ROS production, perpetuate endothelial damage, and cause thrombosis [254,257,258]. C3a and C5a can stimulate the release of IL-6 and TNF from macrophages and other cells expressing the C3a/C5a receptors [259]. SARS-CoV-2 results in overexpression of these receptors, making these pathways more susceptible to activation [260]. In addition, C5a can lead to the release of tissue factor and plasminogen activator inhibitor peptide type I (PAI-1) from endothelial cells [257,260].

Key points:SARS-CoV-2 causes an alteration in the endothelium–pericyte relationship, which leads to the release of inflammatory factors and cell death.SARS-CoV-2, through structural proteins, activates the contact system and the complement pathway. This causes the endothelium to release factors that participate in thrombo-inflammation, such as tissue factor or PAI-1.

#### 4.2.2. The RAAS in Thromboembolism

As mentioned previously, ACE downregulation is linked to the overexpression of ADAM17 that releases ACE2 anchored to the plasma membrane [71], increasing its soluble forms [73,189]. Since ACE2 regulates inflammation, its deregulation increases inflammation. One of the main culprits is ADAM17, which causes the release of proinflammatory factors such as TNF-α, TNF receptor 1 and 2, and IL-6R [185,186]. The release of soluble cytokine receptors and the formation of soluble complexes exacerbates the response through the JAK/STAT3 pathway [133,187]. This inflammatory state contributes to thromboembolic processes [239] and increases angiotensin II levels [195,198], which also contribute to the inflammatory response and coagulation. Angiotensin II is a prothrombotic substance, as it increases the production of PAI-1 in endothelial cells [261]. Thus, an increase in angiotensin II may contribute to the local microthrombus formation in alveolar capillaries that occurs in patients with COVID-19, as fibrin is not degraded by tissue plasminogen activator (tPA) and urokinase-type plasminogen activator (uPA) [262,263]. Finally, it is important to note once again that there is cooperation between the RAAS and the KKS. Due to RAAS deregulation, there is greater production of kinins, which also contribute to perturb haemostasis in patients with COVID-19.

#### 4.2.3. The KKS in Thromboembolism

The main contact system components are FXII, HMWK, and pre-kallikrein, which circulate in the blood in the form of zymogens. They are activated after binding to antigenic molecules of pathogens with high structural variability, such as nucleic acids of microorganisms, NETs, and ferritin [211,264]. Numerous SARS-CoV-2 antigens (e.g., structural proteins, such as S1, N, M, and E) have a high capacity to bind to complement proteins and the contact system, leading to their activation [265,266]. Furthermore, remnants of these antigens after infection may continue to activate these innate immunity pathways, which might contribute to the symptoms in patients with long COVID [267]. The binding of these antigens to proteins of the contact system (HMWK and pre-kallikrein) triggers, through FXII, activation of the intrinsic pathway of blood coagulation [210] and the KKS [212], producing plasma kallikrein and bradykinin that provide feedback to FXII activation [268]. In addition, the contact system can be activated independently of FXII activity, including through angiotensinase C from endothelial cells [206,210] or increased release of proteases (e.g., trypsin) from glandular tissue in response to infection [61]. In COVID-19, the levels and activity of plasma kallikreins and kininogens are increased [216,225,228]. This increased kallikrein activity causes consumption of intrinsic coagulation pathway factors, an increase in the activated partial thromboplastin time [269], conversion of plasminogen to plasmin, and a state of hyperfibrinolysis [236]. Plasmin increases the production of bradykinin, contributing to an inflammatory state [270]. Bradykinin also contributes to this inflammatory and hyperfibrinolytic state through the release of tPA [271], implicating it in the coagulation imbalances that occur in patients with COVID-19 (hyperfibrinolysis or thromboembolism) [272]. In addition, plasmin and FXII can be reciprocally activated [43]. The pathological increase in FXII contributes not only to septic phenomena and blood dyscrasias, but also to fibroblast proliferation and pulmonary fibrosis [273], phenomena that occur in patients with COVID-19 [274].

Key points:Angiotensin II is a factor of thrombo-inflammation in the pulmonary blood capillaries, by increasing the production of PAI-1.Binding of SARS-CoV-2 antigens to the contact system activates the intrinsic coagulation pathway and KKS.The increase in protease activity (e.g., kallikreins or PAR receptors) causes a state of hyperfibrinolysis through the release of tPA. This participates in septic phenomena, blood dyscrasias, fibroblast proliferation, and pulmonary fibrosis.

## 5. Conclusions

SARS-CoV-2 uses plasma membrane and soluble proteases in its infective mechanism. The way it enters the host cell may vary depending on which one is used (e.g., trypsin or cathepsins). In addition, proteases also participate in the replication and maturation of viral proteins, as well as the release of new virions.

Upon viral entry, proteases cleave the S protein to anchor to the host cell via ACE2. This event causes dysregulation of ACE2, increasing its soluble forms and decreasing the one anchored in the cell membrane, a process that is mediated by ADAM17. This has important consequences on inflammation.

The main consequence of ACE2 dysregulation is the increase in the ACE/ACE2 ratio. Because RAAS is an extension of KKS, the release and activity of proteases (e.g., kallikreins) will be unbalanced against the anti-proteases that regulate them. These are going to activate each other in a chain. This imbalance contributes to the infectious and proinflammatory mechanism, as well as to the imbalance of respiratory function. Furthermore, KKS is part of the contact system of innate immunity together with the complement system and activates the intrinsic coagulation pathway.

Angiotensin II and des-Arg^9^-bradykinin are going to be the two main final mediators of this entire chain of events. Among the consequences of this will be the affectation of the endothelium–pericyte relationship, fibroblast proliferation, the death of the vascular endothelium, hyperinflammation, and the release of procoagulant factors, which in the most severe cases cause hyperfibrinolysis, thrombosis, and sepsis. These protease-mediated mechanisms are critical to understanding COVID-19 disease. These mechanisms have not been explained in depth previously.

Aprotinin is a broad-spectrum inhibitor of the most important proteases involved in SARS-CoV-2 infection. We describe its pharmacodynamics, pharmacokinetics, toxicity, and potential for the treatment of various respiratory viruses in a second-part review, entitled “Aprotinin (II): Inhalational Administration for the Treatment of COVID-19 and Other Viral Conditions” [1].

## Figures and Tables

**Figure 1 ijms-25-07553-f001:**
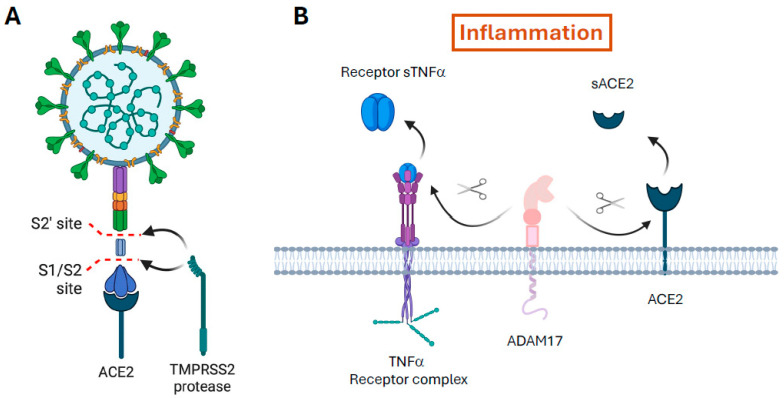
The severe acute respiratory syndrome (SARS-CoV-2) infectious mechanism involves endoproteolysis of the spike (S) protein. (**A**) Representation of S protein cleavage (enlarged in the drawing) by transmembrane serine protease type 2 (TMPRSS2). Cleavage of the furin-like domain produces the active conformation of the S protein, exposing the S1 and S2 subunits. These subunits are responsible for anchoring to angiotensin-converting enzyme 2 (ACE2) located in the plasma membrane of the epithelial cell, and for membrane fusion and viral RNA release. Binding of the virus to ACE2 results in the activation, expression, and release of host proteases (**B**). Metalloproteases, such as a disintegrin and metalloproteinase 17 (ADAM17), cleave the membrane-bound receptors of tumour necrosis factor α (TNF-α) and interleukin 6 (IL-6), as well as ACE2, which are released into the extracellular medium in their respective soluble forms (sTNF receptor 1/2, sIL-6R, and sACE2, respectively). The cleavage and release exacerbate the inflammatory response and is one of the causes of the cytokine storm.

**Figure 2 ijms-25-07553-f002:**
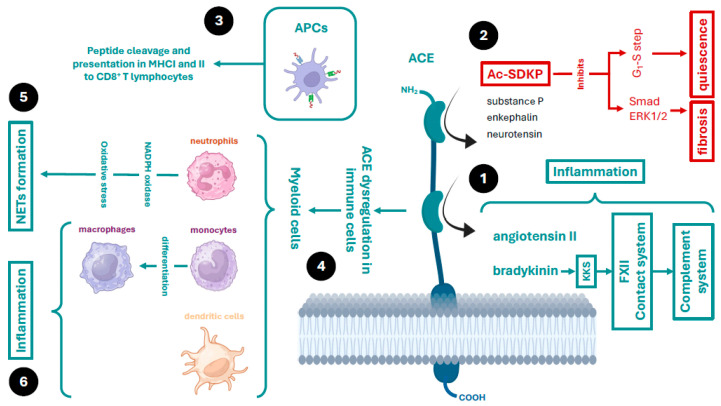
Somatic angiotensin-converting enzyme (ACE) with its two catalytic domains, and its main cellular functions. Somatic ACE has two catalytic domains with different affinities for protein substrates. (1) The carboxyl-terminal domain metabolises bradykinin to inactive peptides. It has greater affinity for bradykinin than it does for angiotensin I. It can regulate angiotensin activity via bradykinin metabolism. Increased angiotensin and bradykinin levels cause inflammation through the angiotensin II receptor type 1 (AT1R) and the bradykinin B1 receptor. In addition, the kallikrein–kinin system (KKS), which includes bradykinin and coagulation factor XII, together with the complement system constitute the contact system of innate immunity. (2) The amino-terminal domain of ACE degrades other peptide transmitters, such as substance P, enkephalins, and neurotensin. However, of great importance is the tetrapeptide N-acetyl-seryl-aspartyl-lysyl-proline (AcSDKP), which maintains myeloid immune cells in a quiescent state. In addition, this peptide blocks small mothers against decapentaplegic (Smad) signal transduction and the extracellular signal-regulated kinase (ERK) 1/2 pathway, which prevents cytokine formation. Thus, AcSDKP has anti-inflammatory and antifibrotic effects and is the counterpart to the effects of angiotensin and bradykinin. (3) ACE cleaves peptides that are 3–42 amino acids for antigen presentation to cluster of differentiation 8 (CD8)^+^ T cells by major histocompatibility complex type I and II (MHC-I and MHC-II) in antigen-presenting cells (APCs). (4) In COVID-19, ACE expression is dysregulated. ACE overexpression in myeloid cells is important for them to acquire immune competence. (5) In neutrophils, ACE actions are mediated by nicotinamide adenine dinucleotide phosphate oxidase (NOX). Overactivity of neutrophils increases oxidative stress and the release and activity of proteases, leading to the production of fibrosis and NETosis. (6) Finally, ACE dysregulation also increases the production of angiotensin II and bradykinin, which through the AT1R and the B1 receptor on myeloid cells act as one of the main chemotactic agents, contributing to cytokine storm and inflammation in COVID-19.

**Figure 3 ijms-25-07553-f003:**
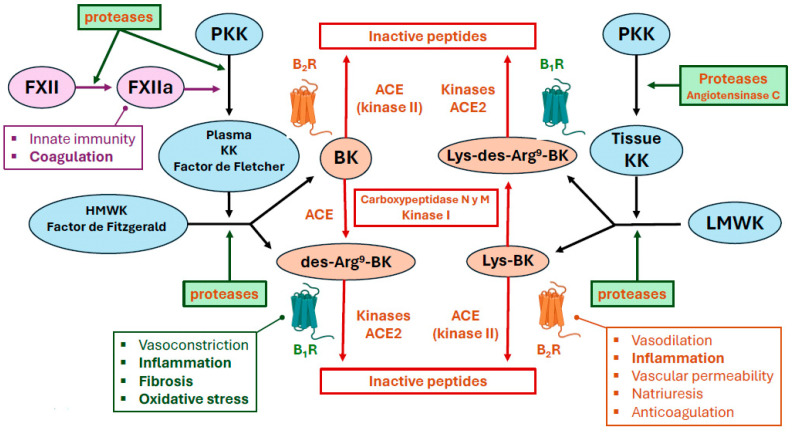
The main actions of the kinin–kallikrein system (KKS). Proteases released from mucosal or myeloid cells provoke activation of factor XII (FXII → FXIIa). FXII can also be activated directly by binding to a viral antigen, as it is part of the contact system of innate immunity. Thus, it can activate the intrinsic coagulation pathway (shown in purple) or, via its serine protease action, plasma pre-kallikrein kininogen (PKK, Fletcher factor) to the kallikrein (KK) zymogen. These zymogens start a chain reaction of cleavage of other peptides to activate their proteolytic action (e.g., plasminogen or protease-activated receptors (PARs)). This process is involved in coagulation, inflammation, and activation of the complement pathway of the adhesion system of innate immunity. In this chain of events, plasma KK also produces high-molecular-weight kininogen (HMWK) and/or Fitzgerald factor, which lead to the production of bradykinin (BK) and des-Arg^9^-BK. PKK can also be released from tissues, which by cleavage of proteases, such as angiotensinase C, produces tissue KK that cleaves low-molecular-weight kininogen (LMWK) to produce Lys-des-Arg^9^-BK and kallidin (also called Lys-BK). In summary, this chain of activation from kininogens to zymogens constitutes the KKS (depicted in blue), and terminates in the production of BK, des-Arg^9^-BK, Lys-BK, and Lys-des-Arg^9^-BK (depicted in orange), each of which have greater or lesser affinity for the bradykinin B1 receptor (B1R) or bradykinin B2 receptor (B2R). Among many actions, B1R and B2R mediate inflammation, fibrosis, and oxidative stress. Furthermore, the importance of these mediators is that they are degraded to inactive peptides by kinases I and II, also known as angiotensin-converting enzyme 1 and 2 (ACE and ACE2, respectively), and by carboxypeptidases. The affinity of these enzymes for these peptides is greater than for angiotensin. Hence, the KKS is intimately related to the renin–angiotensin–aldosterone system.

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
