# Peer review of "Aprotinin (I): Understanding the Role of Host Proteases in COVID-19 and the Importance of Pharmacologically Regulating Their Function"

_ijms, 2024, doi:10.3390/ijms25147553_

Round 1

Reviewer 1 Report

Comments and Suggestions for Authors

The paper is quite well written. The article covers a very interesting and current topic. Nevertheless, in my opinion, some parts need to be improved, I have some comments:

1) Abstract. We aim to elucidate the pathophysiological process of COVID-19 to understand the mechanism of antiviral action of aprotinin (a panprotease inhibitor), and its action on the symptomatic processes of the disease. The conclusions might be beneficial to include a sentence in the abstract that briefly summarizes the key findings of the study. This can provide readers with a quick overview of the research. 

2) 1. Introduction 33 Coronaviruses infect epithelial cells by recognising and binding to certain plasma 34 membrane proteins. One of the most studied is angiotensin-converting enzyme 2 (ACE2). 35 The anchoring mechanism of severe acute respiratory syndrome coronavirus type 2 36 (SARS-CoV-2), which causes coronavirus disease 2019 (COVID-19), to this enzyme has 37 consequences that are fundamental to the cell, and is closely related to the disease it 38 causes. To understand the pathophysiology COVID-19, it is necessary to explain the 39 importance of ACE2, which goes beyond blood pressure control; in this process, ACE2 40 triggers a cascade of events that activate proteolytic pathways. Understanding these 41 mechanisms allows the development of new antiviral drugs, such as aprotinin, which is a 42 broad-spectrum inhibitor of host proteases. Please improve this part and support the information with some references such as:

Quantitative Computed Tomography Lung COVID Scores with Laboratory Markers: Utilization to Predict Rapid Progression and Monitor Longitudinal Changes in Patients with Coronavirus 2019 (COVID-19) Pneumonia. Biomedicines. 2024;12(1):120. Published 2024 Jan 6. doi:10.3390/biomedicines12010120

Radiological-pathological signatures of patients with COVID-19-related pneumomediastinum: is there a role for the Sonic hedgehog and Wnt5a pathways?. ERJ Open Res. 2021;7(3):00346-2021. Published 2021 Aug 23. doi:10.1183/23120541.00346-2021

Quantitative image analysis in COVID-19 acute respiratory distress syndrome: a cohort observational study. F1000Res. 2023;10:1266. Published 2023 Mar 31. doi:10.12688/f1000research.75311.2

3) 2. The SARS-CoV-2 Infectious Process 44 ACE2, the main target to which SARS-CoV-2 anchors in the viral infectious process, 45 is expressed to a greater or lesser extent in the cells of the pulmonary, digestive, renal and 46 vascular endothelium [1,2]. Anchoring and fusion of the viral capsid to the host cell ... Please, add a table with the most important topics and the most important references.

4) Please, add a paragraph and discuss the most important topics 

5) Conclusions. Please, improve this paragraph and underline the novelty of the study.

Comments on the Quality of English Language

Minor changes of English language are required

Author Response

The paper is quite well written. The article covers a very interesting and current topic. Nevertheless, in my opinion, some parts need to be improved, I have some comments:

1) Abstract. We aim to elucidate the pathophysiological process of COVID-19 to understand the mechanism of antiviral action of aprotinin (a panprotease inhibitor), and its action on the symptomatic processes of the disease. The conclusions might be beneficial to include a sentence in the abstract that briefly summarizes the key findings of the study. This can provide readers with a quick overview of the research.

As you suggest, we have changed the last part of the abstract (from line 27 to 32).

2)1. Introduction. Coronaviruses infect epithelial cells by recognising and binding to certain plasma membrane proteins. One of the most studied is angiotensin-converting enzyme 2 (ACE2). The anchoring mechanism of severe acute respiratory syndrome coronavirus type 2 (SARS-CoV-2), which causes coronavirus disease 2019 (COVID-19), to this enzyme has consequences that are fundamental to the cell, and is closely related to the disease it causes. To understand the pathophysiology COVID-19, it is necessary to explain the importance of ACE2, which goes beyond blood pressure control; in this process, ACE2 triggers a cascade of events that activate proteolytic pathways. Understanding these mechanisms allows the development of new antiviral drugs, such as aprotinin, which is a broad-spectrum inhibitor of host proteases. Please improve this part and support the information with some references such as:

- Quantitative Computed Tomography Lung COVID Scores with Laboratory Markers: Utilization to Predict Rapid Progression and Monitor Longitudinal Changes in Patients with Coronavirus 2019 (COVID-19) Pneumonia. Biomedicines. 2024;12(1):120. Published 2024 Jan 6. doi:10.3390/biomedicines12010120

- Radiological-pathological signatures of patients with COVID-19-related pneumomediastinum: is there a role for the Sonic hedgehog and Wnt5a pathways?. ERJ Open Res. 2021;7(3):00346-2021. Published 2021 Aug 23. doi:10.1183/23120541.00346-2021

- Quantitative image analysis in COVID-19 acute respiratory distress syndrome: a cohort observational study. F1000Res. 2023;10:1266. Published 2023 Mar 31. doi:10.12688/f1000research.75311.2

We have rewritten the paragraph you indicate to clarify the objective of the manuscript (from lines 37 to 54). The literature that you mention to us is interesting, but it has no relation to ACE or the action of proteases in COVID-19 disease. The paragraph is only introductory.

3)2. The SARS-CoV-2 Infectious Process ACE2, …the main target to which SARS-CoV-2 anchors in the viral infectious process, is expressed to a greater or lesser extent in the cells of the pulmonary, digestive, renal and vascular endothelium [1,2]. Anchoring and fusion of the viral capsid to the host cell ... Please, add a table with the most important topics and the most important references.

This table (Table 1) with the main mechanisms of viral infection is described in detail in the second part of the manuscript entitled “Aprotinin (II): Inhalational Administration for the Treatment of COVID-19 and Other Viral Conditions” [Int. J. Mol. Sci. 2024, 25(13), 7209; https://doi.org/10.3390/ijms25137209]”. We do not want to repeat the information in this manuscript.

4) Please, add a paragraph and discuss the most important topics

As you suggest, we have added a paragraph at the end of each section highlighting the main points.

5) Conclusions. Please, improve this paragraph and underline the novelty of the study.

We have rewritten the conclusions highlighting the main points and the novelty of the review.

Reviewer 2 Report

Comments and Suggestions for Authors

The author described well on host proteases in COVID-19 in detail. But, I have the following comments on manuscript.

1. Why the author say the title Aprotinin (I) ? (I)?

2. As aprotinin is inhibitor of host protease, many studies used it as treatment in COVID-19 patients. The author mentioned aprotinin and its function. No information on clinical part. It is better to mention it.

3. The function of host protease and host anti-proteases has same role in other coronaviruses?

4. Aprotinin is also ihitr of host protease in human corona virus and other SARS-CoV viruses?

5. The title described importance of pharmacologically regulating their function of Aprotinin and it is better to mention more  how importance of it in treatment side.

6. Is there any side effect of Aprotinin? If so, please discuss it .

Author Response

The author described well on host proteases in COVID-19 in detail. But, I have the following comments on manuscript.”

  1. Why the author say the title Aprotinin (I) ? (I)?

The manuscript has two parts. The first part titled “Aprotinin (I): Understanding the Role of Host Proteases in COVID-19 and the Importance of Pharmacologically Regulating Their Function” is what you are evaluating. It tries to explain the consequences of the activation of proteases in infection with the SARS-CoV-2 virus, in order to understand the action of aprotinin as an antiviral drug, and also to treat the symptomatic processes in COVID-19. This is explained in the second part titled: “Aprotinin (II): Inhalational Administration for the Treatment of COVID-19 and Other Viral Conditions”. I feel that we should have given you this second part of the manuscript so that you could understand the reason for the manuscript.

  1. As aprotinin is inhibitor of host protease, many studies used it as treatment in COVID-19 patients. The author mentioned aprotinin and its function. No information on clinical part. It is better to mention it.

This information is fully reflected in the second part of the manuscript titled “Aprotinin (II): Inhalational Administration for the Treatment of COVID-19 and Other Viral Conditions” as we indicated in your previous question. So that the reader can consult this second part, we have mentioned it in line 50-52 of the manuscript so that it can be consulted in this way.

  1. The function of host protease and host anti-proteases has same role in other coronaviruses?

In general, the symptoms caused by respiratory viruses in the infectious process are due to the release of proteases from epithelial cells. However, there are viruses that, due to their particular infection mechanism through the use of proteases (to cut and activate viral proteins), cause a particular state of hyper-activation and -secretion of proteases. This benefits the virus in its infectious process. In these cases, the need to inhibit its action through drugs such as aprotinin becomes more evident, since it will not only have antiviral activity, but also activity against the symptoms caused by this imbalance in the action of proteases. Viruses that especially take advantage of the action of proteases are those that belong to the family of paramyxoviridae, ortomyxoviridae, retroviridae, herpesviridae, flaviviridae, filoviridae, hepadnaviridae, togaviridae and coronaviridae. This is indicated on line 62-65 of the manuscript.

  1. Aprotinin is also ihitr of host protease in human corona virus and other SARS-CoV viruses?

Yes, it is. As I mentioned in the previous question, the coronaviridae family are viruses that are especially dependent on the use of host proteases in their infection mechanism. Many of the proteases used by coronaviruses are inhibited by aprotinin, therefore, they are greatly affected by the action of this drug. In the second part of the manuscript, Table 1 describes in detail the proteases used by coronaviruses and their ability to inhibit aprotinin, as well as what aspects of the infectious process the virus is affected by this inhibition (anchorage, entry, replication, protein maturation, or virion release).

  1. The title described importance of pharmacologically regulating their function of Aprotinin and it is better to mention more how importance of it in treatment side.

The treatment is reflected in the second part of the manuscript.

  1. Is there any side effect of Aprotinin? If so, please discuss it.

Aprotinin is a very safe drug by inhalation. In clinical trials that have been carried out to date in humans, no major adverse reactions have been described. However, although it is not a particularly immunogenic protein, the possibility of allergic reactions in some patients should not be ruled out. and caution should be taken as it is an exogenous protein. All this is reflected in the second part of the manuscript.

Round 2

Reviewer 1 Report

Comments and Suggestions for Authors

The manuscript has been improved as requested, no further comments.